# Clinical and Histopathological Features of Gelsolin Amyloidosis Associated with a Novel *GSN* Variant p.Glu580Lys

**DOI:** 10.3390/ijms22031084

**Published:** 2021-01-22

**Authors:** Maja Potrč, Marija Volk, Matteo de Rosa, Jože Pižem, Nataša Teran, Helena Jaklič, Aleš Maver, Brigita Drnovšek-Olup, Michela Bollati, Katarina Vogelnik, Alojzija Hočevar, Ana Gornik, Vladimir Pfeifer, Borut Peterlin, Marko Hawlina, Ana Fakin

**Affiliations:** 1Eye Hospital, University Medical Centre Ljubljana, 1000 Ljubljana, Slovenia; maja.potrc@kclj.si (M.P.); brigita.drnovsek@kclj.si (B.D.-O.); ana.gornik@gmail.com (A.G.); vladimir.pfeifer@kclj.si (V.P.); marko.hawlina@gmail.com (M.H.); 2Clinical Institute of Genomic Medicine, University Medical Centre Ljubljana, 1000 Ljubljana, Slovenia; marija.volk@kclj.si (M.V.); natasa.teran@gmail.com (N.T.); helena.jaklic@gmail.com (H.J.); ales.maver@kclj.si (A.M.); borut.peterlin@kclj.si (B.P.); 3Institute of Biophysics, National Research Council, 20133 Milano, Italy; matteo.derosa@ibf.cnr.it (M.d.R.); michela.bollati@guest.unimi.it (M.B.); 4Department of Biosciences, University of Milano, 20133 Milano, Italy; 5Institute of Pathology, Faculty of Medicine, University of Ljubljana, 1000 Ljubljana, Slovenia; joze.pizem@mf.uni-lj.si; 6Department of Neurology, University of Ljubljana, 1000 Ljubljana, Slovenia; vogelnik.kata@gmail.com; 7Department of Rheumatology, University Medical Centre Ljubljana, 1000 Ljubljana, Slovenia; alojzija.hocevar@gmail.com

**Keywords:** gelsolin amyloidosis, Meretoja syndrome, lattice corneal dystrophy, optic neuropathy, *GSN*, cutis laxa, heart arrhythmia, optical coherence tomography

## Abstract

**Simple Summary:**

Gelsolin amyloidosis is a rare autosomal dominant genetic disease, which typically affects the cornea, skin and sometimes other organ systems and is caused by mutations in a gene coding for gelsolin protein (*GSN*). We describe a novel mutation of *GSN* gene, p.Glu580Lys, associated with gelsolin amyloidosis in six members of a two-generation family, who exhibited lattice corneal dystrophy, loose facial skin and irregular heart rhythm. In one patient we reported optic nerve impairment, which is possibly a novel feature associated with gelsolin amyloidosis.

**Abstract:**

Gelsolin amyloidosis typically presents with corneal lattice dystrophy and is most frequently associated with pathogenic *GSN* variant p.Asp214Asn. Here we report clinical and histopathological features of gelsolin amyloidosis associated with a novel *GSN* variant p.Glu580Lys. We studied DNA samples of seven members of a two-generation family. Exome sequencing was performed in the proband, and targeted Sanger sequencing in the others. The heterozygous *GSN* variant p.Glu580Lys was identified in six patients. The patients exhibited corneal dystrophy (5/6), loose skin (5/6) and/or heart arrhythmia (3/6) and one presented with bilateral optic neuropathy. The impact of the mutation on the protein structure was evaluated in silico. The substitution is located in the fifth domain of gelsolin protein, homologous to the second domain harboring the most common pathogenic variant p.Asp214Asn. Structural investigation revealed that the mutation might affect protein folding. Histopathological analysis showed amyloid deposits in the skin. The p.Glu580Lys is associated with corneal dystrophy, strengthening the association of the fifth domain of gelsolin protein with the typical amyloidosis phenotype. Furthermore, optic neuropathy may be related to the disease and is essential to identify before discussing corneal transplantation.

## 1. Introduction

Gelsolin amyloidosis (also known as AGel, familial amyloidosis of Finnish type or Meretoja syndrome) is a form of autosomal dominant systemic amyloidosis associated with pathogenic mutations in the *GSN* gene, encoding the multidomain protein gelsolin [1]. Gelsolin is an actin-binding protein involved in cell movement, cytokinesis, and apoptosis. It consist of six homologous domains, G1–G6. At resting intracellular Ca ion levels, the arrangement of the domains cover the actin-binding sites. With elevation of Ca(2+) concentration the structure of domains’ complex changes and allows actin binding [2].

Only a few hundred cases of patients with gelsolin amyloidosis have been described in the literature, mostly of Finnish origin harboring the pathogenic variant p.Asp214Asn in the domain G2 [3]. For a long time, gelsolin amyloidosis has been associated with only mutations of the residue 214, the aforementioned p.Asp214Asn and p.Asp214Tyr, also known as the Danish variant [4,5]. Both mutations were shown to compromise calcium binding [6,7].

Most frequently reported clinical features include lattice corneal dystrophy, loose skin (cutis laxa), and cranial nerve involvement, most frequently of the facial and trigeminal nerves [3,8]. Other reported features include carpal tunnel syndrome, proteinuria and renal failure, heart arrhythmia and atrioventricular block [9]. Among the ocular signs, dry eye, Meibomian gland insufficiency, photophobia, early onset cataract, and secondary open angle glaucoma have been reported [10]. Autopsy examination of the eyes of patients with gelsolin amyloidosis revealed ocular amyloid deposits in the conjunctiva, sclera, perineurium of the ciliary nerves, walls of the ciliary vessels, optic nerve sheaths, stroma of the ciliary body and along the choriocapillaris [11,12]. In recent years, cases harboring mutations of other residues of the *GSN* have also been reported. These substitutions reside in other domains of the protein and, in some cases, their clinical presentation diverge from the classical one [13,14,15,16,17,18]. Here we report clinical and histopathological features of gelsolin amyloidosis associated with a novel *GSN* variant c.1738G>A (p.Glu580Lys) and review phenotypes associated with all known *GSN* variants to date.

## 2. Results

### 2.1. Clinical Presentation

The proband (II-1) was a 63-year old Caucasian woman, who had been followed due to lattice corneal dystrophy since the age of 56. At the age of 60, her visual acuities were 200/200 on the right eye (RE) and 20/200 on the left eye (LE) and she was awaiting LE corneal transplantation. At the age of 63, she presented with a 3-week history of blurred vision on her RE that occurred five days after knee surgery, following which she had a period of hypotension. Her best corrected visual acuity on the RE was 20/200, there was a reduction of her color vision, and positive relative afferent pupillary defect (RAPD). Goldmann kinetic perimetry demonstrated bilateral constriction of the visual fields with absent isopters V1 and V2 but no localized scotoma (Appendix A). Slit lamp examination revealed bilateral linear white stromal corneal deposits (Figure 1A,C) and mild lens opacities, bilaterally tortuous retinal vessels, optic disc edema on RE and pale optic disc on the LE (Figure 2A). She had loose facial skin (cutis laxa) with most notable dermatochalasis of the upper and lower eyelids (Figure 3). Family history revealed that her late mother and three siblings, who were living abroad, had similar corneal problems and facial appearance; and her daughter also had corneal problems (Figure 1B,D, Table 1).

Extensive evaluation was performed due to the right eye acute optic nerve edema. Optical coherence tomography (OCT) of the RE showed thickening of the retinal nerve fiber layer (RNFL) around the optic nerve and thinning of the ganglion cell layer in the macula, while OCT of the LE showed thinning of the RNFL around the optic nerve (Figure 2A). The LE macular OCT showed no gross abnormality but was of too poor quality for quantitative analysis due to the corneal opacities. Fluorescein angiography revealed leakage from the RE optic disc (Appendix A). Laboratory work-up showed normal complete blood count and moderately increased erythrocyte sedimentation rate (ESR) (64 mm/h) and C-reactive protein (CRP) (10 mg/L). Renal function tests displayed normal creatinine and glomerular filtration rate values. Giant cell arteritis was excluded with the ultrasound of the temporal arteries (Appendix A) and the absence of other clinical symptoms such as headache or jaw claudication. Computed tomography of the brain excluded compressive lesions but showed multiple calcifications in the brain nuclei (Appendix A), suggesting possible Morbus Fahr (primary familial brain calcification) or neurocysticercosis. The latter was excluded with negative serology for Taenia solium. Serology tests for Treponema pallidum, Lyme disease, and Toxoplasma gondii were also negative. Expanding intracranial lesions were excluded with the MRI of the brain, however, white matter punctate lesions were detected in the frontal lobes. On follow-up exam after 7 weeks, the visual acuity was counting fingers at 1 m RE and 20/125 LE, and there was bilateral atrophy of the optic nerve (Figure 2B). Neurological examination revealed signs of mild sensory ataxia and symptoms suggestive of carpal tunnel syndrome (CTS). She is awaiting neurophysiology studies to help localize the lesion responsible for sensory ataxia and to confirm the diagnosis of CTS. No cranial nerve involvement other than optic neuropathy was documented.

The proband’s daughter (III-1) presented at the emergency ophthalmology department complaining of blurred vision and retrobulbar pain on her LE at the age of 40, when lattice corneal dystrophy was first observed. Six years later, after receiving her mother’s genetic results, we invited her for an ophthalmological examination and genetic testing. At that time, she complained of glare and dry eye sensation as well as little, short lines in visual field of both eyes. Visual acuity was 20/25 on her RE and 20/20 on her LE. Octopus G2 top static perimetry revealed centrally decreased sensitivity bilaterally. There was no obvious dermatochalasis or other skin laxity, while bilateral corneal lattice dystrophy was seen on the slit lamp (Figure 1B,D). OCT of the optic nerve head was normal, however, the ganglion cell layer was thinned on the macular OCT (Figure 2C). Magnetic resonance imaging revealed left frontal focal cortical encephalomalacia due to a traumatic head injury she suffered 20 years prior. Neurological examination showed blepharospasm, oromandibular dystonia and torticollis to the right with ’no-no’ type tremor of the head. In addition, increased muscle tone in the right leg, brisk tendon reflexes and bilateral extensor plantar response were noted and attributed to the old injury. No cranial nerve involvement was observed.

Four affected relatives who lived abroad (aged 39–72 years) filled out a questionnaire regarding their medical history (Appendix A). All reported skin laxity (Appendix A), three reported corneal lattice dystrophy, three had heart arrhythmia treated with implantable cardioverter defibrillator and two reported renal involvement. None of them reported cranial palsy. The clinical presentation of all studied family members is summarized in Table 1.

### 2.2. Molecular Analysis

Exome sequencing in the proband revealed a novel heterozygous variant c.1738G>A (p.Glu580Lys) in the *GSN* gene (NM_000177.5, Appendix A). The variant is absent from the controls (GnomAD. Available online: GnomAD.broadinstitute.org (accessed on 5 June 2020)) and the majority of in silico predictors indicate its damaging effect (including MetaSVM, REVEL and CADD meta-predictors). Family genetic studies revealed the presence of the GSN variant in five relatives and confirmed the co-segregation of the variant with the disease status. Based on the ACMG guidelines for variant classification [19], adapted in accordance with ACGS recommendations (Association for Clinical Genomic Science. Available online: https://www.acgs.uk.com/quality/best-practice-guidelines/ (accessed on 16 December 2020)), we classified the variant as likely pathogenic. The evidence for this classification was based on variant’s absence in the general population (PM2), strong segregation evidence (PP1_STR), theoretical predictions of pathogenicity (PP3), the results of a protein modeling study performed for this variant (PM1_SUP) and the consistency of the finding with the clinical presentation observed in the patients with this variant (PP4).

To evaluate the impact of the mutation on the structure of the protein, we built an in silico model of the p.Glu580Lys variant (Figure 4). The p.Glu580 residue belongs to the G5 protein domain and locates at the interface between this and the fourth domain. In the wild-type protein, it plays a crucial role for the maintenance of the G4–G5 interaction, forming a salt-bridge with p.Arg564 and H-bonds with other residues of the G4. According to the model, in the p.Glu580Lys variant, the mutated residue is not able to maintain most of these interactions, with a significant loss of connectivity and stability. Moreover, the Glu to Lys substitution leads to a local inversion of the charge, from negatively to positively charged, in an area of the protein where other positively charged residues cluster, likely causing electrostatic repulsion.

### 2.3. Pathological Findings

Epidermis and dermis of the eyelid skin were atrophic. There was prominent amyloid deposition (Figure 5) around hair follicles; sebaceous and eccrine glands, multifocally at the epidermal basal membrane, within vessel walls (small arteries, arterioles and focally venules), within skeletal muscle cells (predominantly at the level of sarcolemma) and focally around collagen fibers. The amyloid deposits showed positivity for Congo red stain (Figure 5A–D) including characteristic green birefringence and were highlighted by thioflavin T stain (Figure 5E,F). Electron microscopic examination showed 6–10 nm thick fibrillary deposits, characteristic of amyloid (Figure 5). Congo red and thioflavin T stain did not show amyloid deposition in the lens tissue (anterior capsule, lens epithelial cells or fragments of the lens cortex).

### 2.4. Review of the Phenotypes Associated with GSN Mutations

Ten *GSN* mutations, including ours, and their associated phenotypes are summarized in Table 2 and their location in *GSN* protein is shown in Figure 6. The typical phenotype was also originally described in families harboring the common pathogenic variant p.Asp214Asn, its hallmark being corneal lattice dystrophy. This symptom was observed in association with three other pathogenic variants, namely p.Asp214Tyr, p.Met544Arg and p.Glu580Lys (present report). All these substitutions reside either in the second domain of the protein or at the interface between the fourth and fifth protein domains. Five other *GSN* mutations were described in association with other phenotypes, such as renal amyloidosis (p.Gly194Arg and p.Asn211Lys), dermatomyositis (p.Pro459Arg) and seizures (p.Ala7fs).

## 3. Discussion

This study reports clinical and histopathological features of a novel mutation in *GSN*, p.Glu580Lys. Histopathological analysis revealed amyloid deposition in the hair follicles; sebaceous and eccrine glands, multifocally at the epidermal basal membrane, within vessel walls, within skeletal muscle cells and focally around collagen fibers. Some clinical characteristics of the six affected members of a two-generation family were typical of gelsolin amyloidosis, with the most frequent features being corneal lattice dystrophy, cutis laxa and heart arrhythmia. Contrary to the typical phenotype, facial nerve palsy was not frequent and the proband exhibited consecutive optic neuropathy, not yet reported in gelsolin amyloidosis.

### 3.1. Optic Nerve Involvement

Optic neuropathy has not yet been reported as a feature of gelsolin amyloidosis but was a prominent clinical feature in the patient II-1, who presented to the Eye Hospital at the age of 63 years with optic nerve edema and contralateral optic nerve atrophy. The extensive evaluation excluded frequent infectious and autoimmune diseases, giant cell arteritis and compressive brain lesions. Considering the sudden onset and non-progressive nature, the cause was likely ischemia, although the constricted visual field was not typical. The optic nerve atrophy on the other eye suggested a similar prior event on the LE, possibly unnoticed due to the opacities in the cornea. On the other hand, optic neuropathy could also result from amyloid deposition of optic nerve sheaths, however, the infiltrative pathogenesis would suggest a more gradual visual deterioration.

Optic neuropathy or other optic nerve involvement has not been previously reported in association with gelsolin amyloidosis, however, bilateral consecutive anterior ischemic optic neuropathy has been documented in patients with the sporadic light-chain amyloidosis [21,22]. The possible amyloid mechanisms underlying optic nerve edema include direct optic nerve infiltration with amyloid or amyloid deposition in the arterial walls perfusing the optic nerve, causing stenosis and reduced adaptability to systemic blood pressure changes. Previous histopathological studies in *GSN* patients have indeed shown deposition of gelsolin fragments with amyloidogenic properties in the optic nerve sheaths, the walls of ciliary vessels and along the choriocapillaris [9,11,12]. The combination of symptoms and signs with sudden visual deterioration, which is atypical for infiltrative neuropathy, and the absence of the altitudinal visual field defect typical for ischemic optic neuropathy suggest a combination of both mechanisms—infiltrative and ischemic.

This case illustrates the importance of optic nerve evaluation in patients with lattice corneal dystrophy, especially in those awaiting corneal transplant surgery. If the media are too opaque for reliable optic nerve evaluation on the slit lamp, OCT can be used to measure RNFL and macular GCL thickness. Interestingly, the brain lesions in our patient also remained unexplained and may be associated with the syndrome. We suggest that MRI is performed in patients with this syndrome to determine whether the occurrence of such lesions is more frequent.

### 3.2. Pathogenetic Mechanisms and Genotype-Phenotype Correlations

Gelsolin is a relatively large protein which harbors six homologous domains, sequentially named G1 to G6 [23]. *GSN* originated by multiple gene duplication events, duplication of a triplet being the most recent. As a consequence, all gelsolin domains are homologue but highest sequence and structural similarity is observed between these couples: G1 and G4, G2 and G5, G3 and G6 [23]. For decades, G2, which hosts the classic p.Asp214Asn or p.Asp214Tyr mutations, was thought to be the only domain associated with gelsolin amyloidosis. Residue number 214 is a part of a cluster of amino acids chelating a calcium ion. Both mutations of this residue compromise calcium binding [6,7], leading to a destabilization of the protein that becomes susceptible to aberrant proteolysis. These mutations increase the conformational flexibility of G2 and, as a consequence, the exposure of a stretch of the domain that is aberrantly cleaved by furin and matrix-metalloproteases with these proteolytic events leading to the production of amyloidogenic peptides [6,24,25]. Recently described mutations in G4 [16,17] and G5 domains [18], however, suggest an alternative and furin-independent pathway of gelsolin aggregation. The in silico model of the p.Glu580Lys variant predicts a net loss of connectivity and electrostatic repulsion caused by the Lys substitution, which might cause protein destabilization and misfolding, which are features often associated to pathological aggregation and aberrant proteolysis [26].

In recent years, several *GSN* mutations have been discovered and reported in patients with typical and atypical clinical presentations (Table 2). The classic phenotype that has originally been described in the large families with p.Asp214Asn mutation [3] has previously been observed only in patients harboring another mutation of the same residue (p.Asp214Tyr) [20] suggesting that the residue was phenotype-specific. This has been challenged by a recent publication of Cabral-Macias et al. in 2020, of a novel mutation p.Met544Arg, located in the G4:G5 interface, also causing a typical gelsolin phenotype, including the pathognomonic corneal lattice dystrophy [17]. The mutation p.Glu580Lys described in the present study is also located in the G4:G5 interface and associated with the typical phenotype and supports the observation that mutations at various locations are able to produce classic phenotype of gelsolin amyloidosis and a common pathogenetic pathway. Reports of cases harboring other missense mutations in the domains G2 (p.Gly194Arg, p.Asn211Lys), G4 (p.Pro459Arg) and the G4:G5 interface (p.Ala578Pro) described more localized phenotypes including only renal, skin and/or heart disease without corneal lattice dystrophy [13,14,15,16,17,18]. These clinical presentations were still within the classic phenotypic spectrum and most reports included only a small number of cases, therefore, further studies are needed to conclude whether they cause identical or a limited/overlapping phenotype. The disease pathogenesis may also differ. For example, two of the mutations in the G2 domain, p.Gly194Arg and p.Asn211Lys, do not impair calcium binding [24,25] and patients carrying these mutations only present with renal complications [14,15]. Nevertheless, in comparison to missense mutations mentioned above, the frame-shifting mutation p.Ala34fs produced a notably different phenotype with none of the classic features but with severe brain involvement including seizures and brain lesions [13], possibly related to haploinsufficiency. These observations suggest that more basic research and clinical studies are required to decipher and ultimately cure this rare yet disabling disease.

## 4. Materials and Methods

### 4.1. Patients

Seven members of a two-generation family originating from Herzegovina region of Bosna and Herzegovina were ascertained for the study (pedigree shown in Figure 7). The proband and her daughter (II-1 and III-1) underwent a detailed ophthalmological exam while five relatives, who were unavailable for examination due to living abroad, filled out a clinical questionnaire regarding their medical history (Appendix A) and provided saliva samples for genetic analysis.

All participants signed a written informed consent prior to examination.

### 4.2. Clinical Examination

Clinical examination in the patients II-1 and III-1 included visual acuity, visual field (manual Goldmann kinetic perimetry (Haag Streit, Berne, Switzerland) or Octopus static perimetry (Octopus 101, program G2, Interzeag AG, Schlieren, Switzerland), slit lamp examination, optical coherence imaging of the macula and the optic nerve (swept source optical coherence tomography; SS-OCT; Triton^tm^, Topcon, Tokyo, Japan), and neurological exam. The patient II-1 also underwent an additional work-up due to acute optic nerve edema (described in detail in Results).

### 4.3. Genetic and Bioinformatic Analysis

Genetic analyses were performed in seven family members. Genomic DNA was extracted from blood or saliva samples according to the standard procedure. Sequencing of the defined clinical target was performed using next-generation sequencing on the isolated DNA sample of the proband, while in other family members, segregation analysis of identified variant was carried out using Sanger sequencing. Briefly, the fragmentation and enrichment of the isolated DNA sample were performed according to the Illumina Nextera Coding Exome capture protocol (Illumina, USA), with subsequent sequencing on Illumina NextSeq 550 in 2 × 100 cycles. After duplicates were removed, the alignment of reads to UCSC hg19 reference assembly was done using BWA algorithm (v0.6.3) and variant calling was done using GATK framework (v2.8). Only variants exceeding the quality score of 30.0 and depth of 5 were used for down-stream analyses. Variant annotation was performed using ANNOVAR and snpEff algorithms, with pathogenicity predictions in dbNSFPv2 database. Reference gene models and transcript sequences are based on RefSeq database. Structural variants were assessed using CONIFER v0.2.2 algorithm. Variants with population frequency exceeding 1% in gnomAD, synonymous variants, intronic variants and variants outside the clinical target were filtered out during analyses. An in-house pipeline was used for bioinformatic analyses of exome sequencing data, in accordance with GATK best practice recommendations [27]. The interpretation of sequence variants was based on ACMG/AMP standards and guidelines [19]. Sequencing the DNA sample, we reached median coverage of 67× and covered over 99.9% targeted regions with minimum 10× depth of coverage [28]. Segregation analysis in the family members was performed using targeted Sanger sequencing. Primer sequences are available upon request.

### 4.4. In Silico Analysis

The impact of the novel mutation on the structure of the gelsolin protein was evaluated in silico. Structure of the p.Glu580Lys was obtained by in silico mutagenesis of the wild-type protein (pdb id 3FFN [29], followed by energy minimization to resolve unfavorable interactions and clashes [30]. PyMOL (the PyMOL Molecular Graphics System, Version 2.0 Schrödinger, LLC.) was used for the analysis of the interactions and preparation of the figure.

### 4.5. Pathological Analysis

Eyelid skin tissue was obtained during blepharoplasty for upper eyelid dermatochalasis (performed by B.D.-O., Figure 3) and lens tissue was obtained during cataract surgery (performed by V.P.) of the patient II-1. The tissue samples were fixed in formalin and embedded in paraffin. Hematoxylin and eosin, Congo red and thioflavin T stains were performed. Electron microscopic examination of the skin sample was performed on the formalin-fixed tissue.

Congo red staining was performed automatically in Ventana Benchmark Special Stains stainer with Congo Red Staining Kit (Ventana Medical Systems Inc., Tucson, AZ, USA). For thioflavin T staining, slides were incubated in 1% working solution of thioflavin T for 7 min (Sigma Aldrich, Darmstadt, Germany), rinsed in deionized water and then kept in 1% CH3COOH for 20 min. Afterwards, the slides were rinsed and coverslipped directly from deionized water with Dako Fluorescence Mounting Medium (DAKO, Glostrup, Denmark).

### 4.6. Review of the Literature

Studies reporting *GSN* mutations accessible in the PubMed database in the period from 1969–2020 were reviewed. The nomenclature of the reported mutations was annotated according to the originally transcribed and the cleaved plasma protein [31] to avoid confusion in the nomenclature. The phenotypes were evaluated in order to determine whether they included the typical corneal lattice dystrophy and/or other clinical features.

Research protocols adhered to the tenets of the Declaration of Helsinki. The informed consent was acquired from the patients.

## 5. Conclusions

The novel *GSN* variant p.Glu580Lys is associated with the typical gelsolin amyloidosis phenotype, including its hallmark, the corneal lattice dystrophy. Furthermore, optic neuropathy may be associated with the disease and is important to identify before corneal transplantation.

Although the novel pathogenic variant shares a similar clinical picture to FAF, the underlying molecular mechanism might be different. Glu580Lys mutation, as many others reviewed in this manuscript, localizes far from the second domain and unlikely leads to its destabilization and the exposure of the aberrant furin cleavage site. Recent molecular studies [28] suggested an alternative, proteolysis-independent mechanism, but they await in vivo validation.

## Figures and Tables

**Figure 1 ijms-22-01084-f001:**
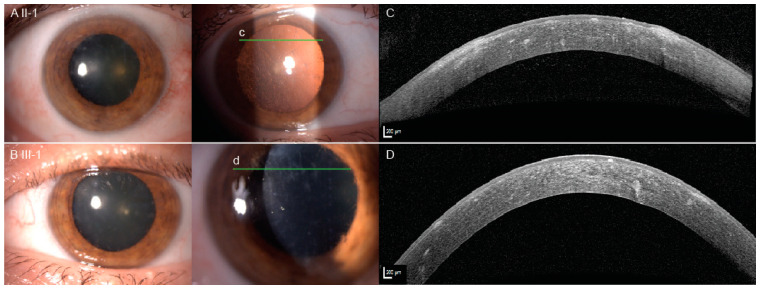
Slit lamp pictures (**A**,**B**) and the anterior segment optical coherence tomography (**C**,**D**) of the patients II-1 and III-1. The lines c and d mark the location of the optical coherence tomography (OCT) scans. Note the thick anterior and midstromal filaments on the OCT.

**Figure 2 ijms-22-01084-f002:**
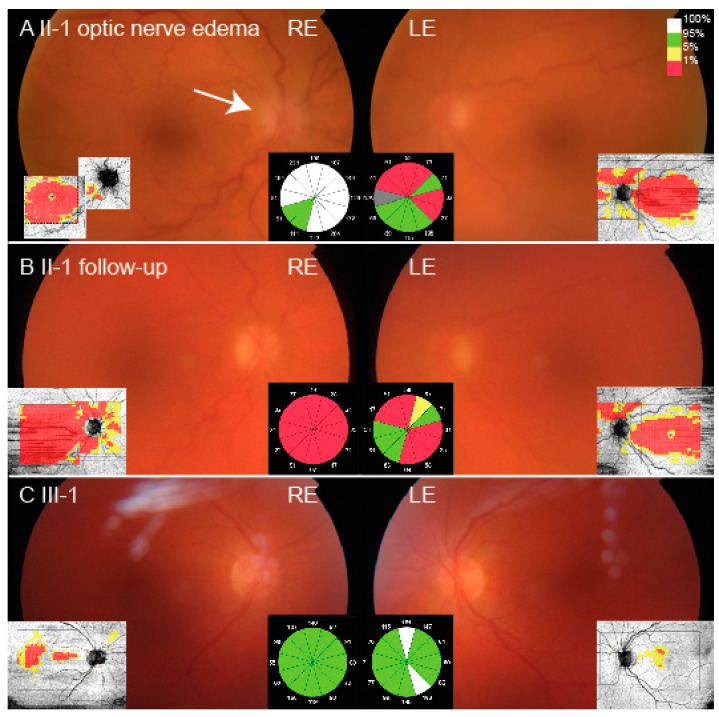
Fundus images of the patient II-1 (**A**,**B**) and III-1 (**C**). Note the optic nerve edema (white arrow) and vascular tortuosity on presentation in patient II-1 (**A**) which resolved on follow-up 7 weeks later (**B**). The circles placed in the middle represent the thickness of the peripapillary retinal nerve fiber layer where the average thickness of each sector is noted with number in micrometers. The ganglion cell layer thickness in the macula is shown on the side of each fundus image. The different colors represent the comparison of measured thickness with the normative data (with the red color representing thickness observed in less than 1% of the population (see color coded legend placed at the top right corner of the figure)). N/A = not available.

**Figure 3 ijms-22-01084-f003:**
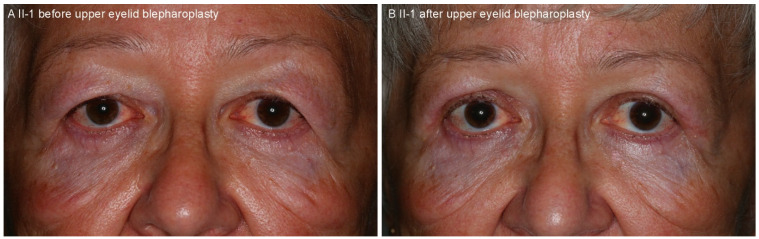
The facial appearance of the patient II-1 before (**A**) and after (**B**) the upper eyelid blepharoplasty. Note the loose skin (cutis laxa) of the upper and lower eyelids, characteristic for gelsolin amyloidosis.

**Figure 4 ijms-22-01084-f004:**
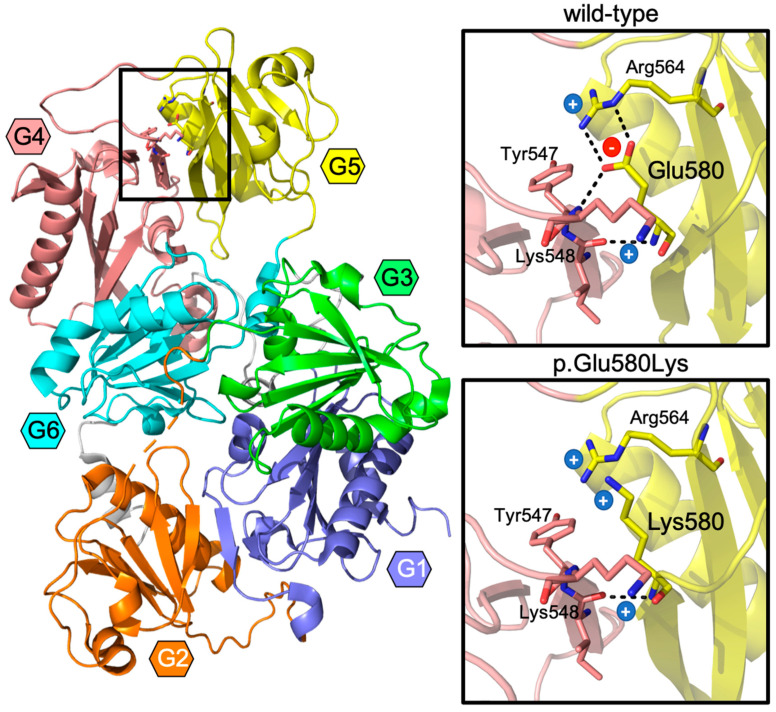
The impact of the mutation on the structure of the protein. Gelsolin protein is organized in 6 homologous domains, labeled G1 to G6 and uniquely colored in the figure. The p.Glu580Lys substitution locates at the G4–G5 interface, highlighted by the frame. A close-up view of this area in the wild-type and mutated protein is reported in the insets, where residue 580 and those interacting are represented as sticks and labeled. Electrostatic interactions and H-bonds are displayed as dashed lines; positively and negatively charged residues are marked by a blue (+) and red (−) symbol, respectively.

**Figure 5 ijms-22-01084-f005:**
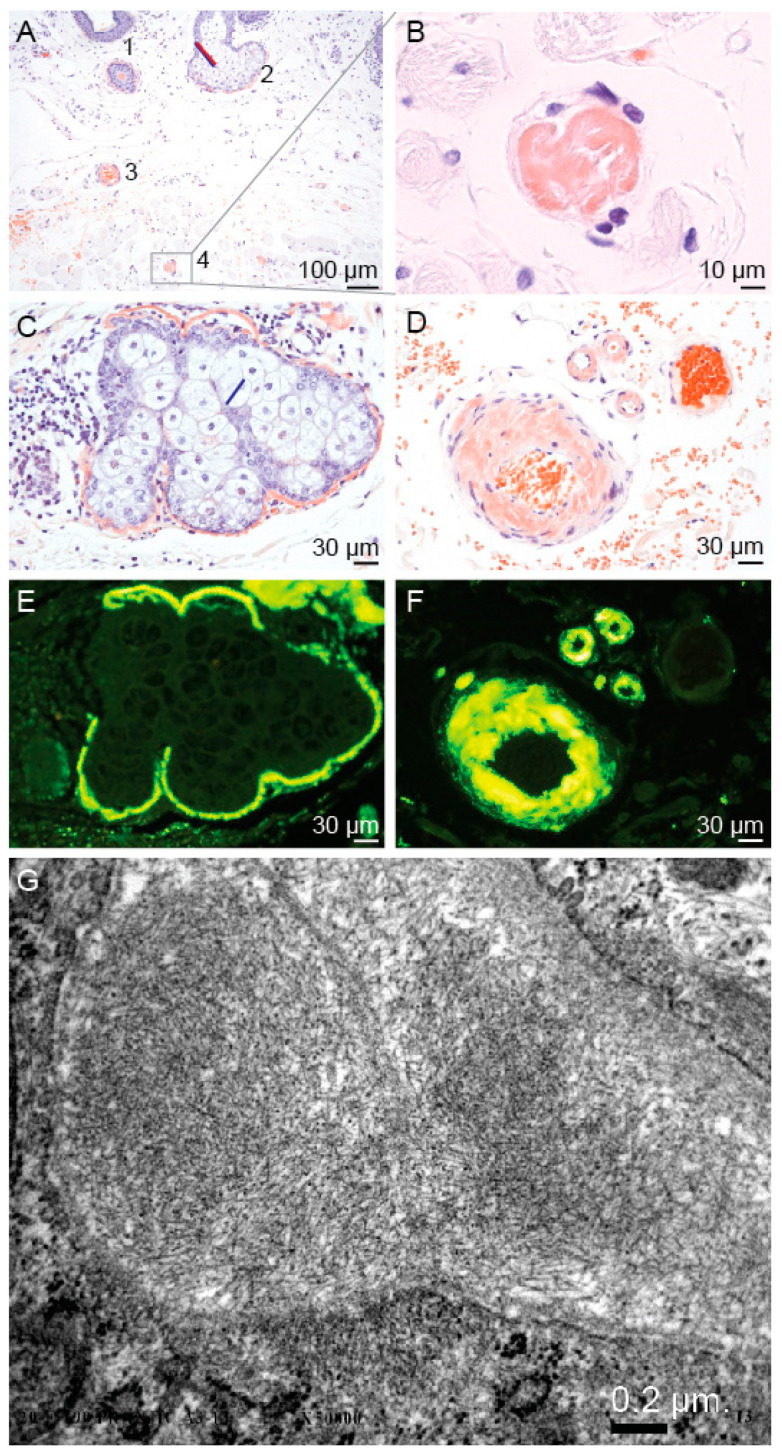
Histopathological analysis of the right upper eyelid blepharoplasty specimen of the patient II-1. A Congo red stain shows amyloid deposition at the basal membrane of a hair follicle (**1**) and a sebaceous gland (**2**), within a vessel wall (**3**) and within a skeletal muscle cell (**4**) (**A**). Higher magnifications showing amyloid deposition within a skeletal muscle cell (**B**) (higher magnification of the area depicted in (**A**), around a sebaceous gland (**C**) and within vessel walls of a small artery and arterioles (**D**). Amyloid deposits highlighted by thioflavin T stain around a sebaceous gland (**E**) and within vessel walls (**F**). Areas in (**E**) and (**F**) correspond to (**C**) and (**D**), respectively. Electron microscopy (**G**) shows amyloid fibrils within the vessel wall, beneath an endothelial cell (upper right).

**Figure 6 ijms-22-01084-f006:**
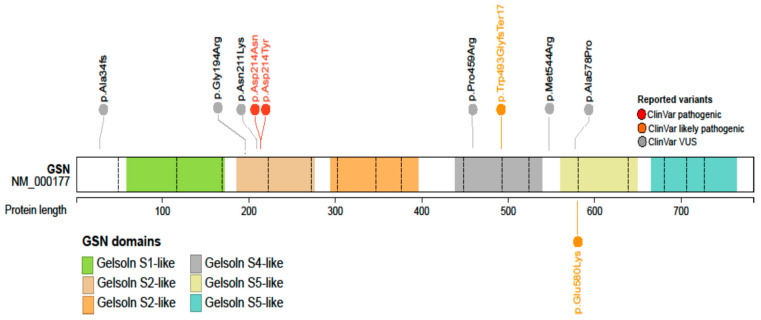
*GSN* variants with their location in the GSN protein and their pathogenicity designation according to the ClinVar database (Available online: ncbi.nlm.nih.gov/clinvar/ (accessed on 5 December 2020)). The phenotypes that were reported in association with these variants are listed in Table 2. Note the mutation p.Trp493GlyfsTer17 that was designated as likely pathogenic but not yet described in patients in the literature.

**Figure 7 ijms-22-01084-f007:**
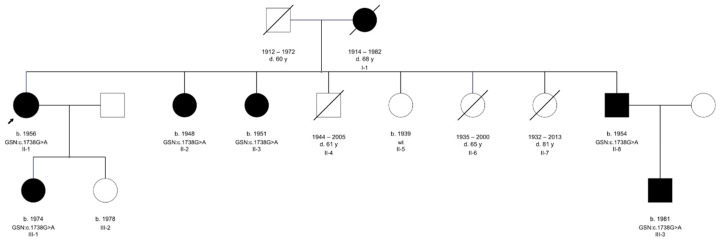
Three-generation family tree of seven members, six of them affected, is presented. Proband is indicated with an arrow. The genotypes of available family members are subscribed indicating the cosegregation of the identified *GSN* variant with the phenotypic features.

**Table 1 ijms-22-01084-t001:** The clinical presentation of the studied family members.

ID	Sex	DOB (Age)	Genotype	Clinical Presentation					
				Corneal Lattice Dystrophy	Loose Skin	Cranial Neuropathy	Heart Arrhythmia	Renal Involvement	Other Phenotypic Features
II-1 *	F	1956 (64 y)	c.1738G>A (p.Glu580Lys)	+(diagnosed at 56 y)	+(dermatochalasis)	+(optic neuropathy)	-		cataract, brain calcifications, sensory ataxia, carpal tunnel syndrome, white matter lesions
II-2	F	1948 (72 y)	c.1738G>A (p.Glu580Lys)	+(diagnosed at 50 y)	+(not specified)	-	ICD (63 y)	-	diabetes mellitus, insomnia
II-3	F	1951 (69 y)	c.1738G>A (p.Glu580Lys)	+(diagnosed at 44 y)	+(dermatochalasis)	-	ICD (65 y)	urolithiasis, renal cysts (diagnosed at 40 y)	angina pectoris, coronary angioplasty and stenting (50 y), hypertension
II-8	M	1954 (66 y)	c.1738G>A (p.Glu580Lys)	+(diagnosed at 60 y)	+(not specified)	-	ICD (62 y)	-	cataract, open angle glaucoma
III-1 *	F	1974 (46 y)	c.1738G>A (p.Glu580Lys)	+(diagnosed at 40 years)	-	-	-	-	torticollis, tremor with sensory tic, blepharospasm, oromandibular dystonia
III-3	M	1981 (39 y)	c.1738G>A (p.Glu580Lys)	-	+(dermatochalasis)	-	-	urolithiasis (diagnosed at 28 y)	
II-5	F	1939 (81 y)	Wild type	-	-	-	-	-	hypertension, dilated cardiomyopathy, aortic valve stenosis with mechanical valve replacement

* The individuals marked with an asterisk (mother and daughter) were examined at the University Medical Centre Ljubljana while others filled out a questionnaire regarding their medical history. ICD = implantable cardioverter defibrillator.

**Table 2 ijms-22-01084-t002:** *GSN* mutations and associated phenotypes.

Base Change (NM_000177.4)	Amino Acid Change	Mature Plasma Protein Numbering (Devoid of the 27-aa Signal Peptide)	GnomAD Allele Frequency	Polyphen (Uniprot ID P06396)	Corneal Lattice Dystrophy	Other Clinical Features	Reference	Localization: Protein Domain (G1-6) *	Protein Destabilization	Susceptibility to Furin Proteolysis	Molecular Reference
c.100dupG	p.Ala34fs	p.Ala7fs	0	/	not reported	seizures, brain lesions	Feng et al. 2018 [13]	/	N/A	N/A	
c.580G>A	p.Gly194Arg	p.Gly167Arg	0.000039	Probably damaging	not reported	renal involvement	Sethi et al. 2013 [14]	G2	+	+	Bonì 2018
c.633C>A	p.Asn211Lys	p.Asn184Lys	0	Probably damaging	not reported	renal involvement	Efebera et al. 2014 [15]	G2	+	+	Bonì 2016
c.640G>A	p.Asp214Asn+	p.Asp187Asn^#^	0.000007	Probably damaging	yes	cutis laxa, cranial neuropathy, heart arrhythmia, renal involvement	Meretoja, Ann Clin Res. 1969 [3]	G2 (Ca-binding site)	+	+	Isaacson 1999
c.640G>T	p.Asp214Tyr	p.Asp187Tyr	0	Probably damaging	yes	cutis laxa, cranial neuropathy	de la Chapelle et al. 1992 [20]	G2 (Ca-binding site)	+	+	Isaacson 1999
c.1375C>G	p.Pro459Arg	p.Pro432Arg	0.000004	Possibly damaging	not reported	dermatomyositis -like	Oregel et al. 2018 [16]	G4	N/A	N/A	
c.1631T>G	p.Met544Arg	p.Met517Arg	0	Probably damaging	yes	cutis laxa, peripheral neuropathy	Cabral-Macias et al. 2020 [17]	G4 (G4-G5 interface)	N/A	N/A	
c.1476del	p.Trp493GlyfsTer17	p.Trp466fsTer17	0	Probably damaging (GERP score 5.25)	N/A	N/A	ClinVar ID: 493491 https://www.ncbi.nlm.nih.gov/clinvar/(accessed on 5 December 2020)	G4	N/A	N/A	
c.1732G>C	p.Ala578Pro	p.Ala551Pro	0	Probably damaging	not reported	cardiac involvement (additional mutation in *TTR*)	Sridharan et al. 2018 [18]	G5 (G4:G5 interface)	N/A	N/A	
c.1738G>A	p.Glu580Lys	p.Glu553Lys	0	Probably damaging	yes	cutis laxa, cranial neuropathy including the optic nerve, cardiac involvement	This report	G5 (G4:G5 interface)	N/A	N/A	

# the most frequent pathogenic variant. * domain nomenclature is from Burtnick et al. 1997. N/A = not available.

## Data Availability

The data presented in this study are available on request from the corresponding author. The data are not publicly available due to personal data protection.

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
