# Peer review of "Clinical and Histopathological Features of Gelsolin Amyloidosis Associated with a Novel GSN Variant p.Glu580Lys"

_ijms, 2021, doi:10.3390/ijms22031084_

Round 1

Reviewer 1 Report

The paper describes a novel gelsolin variant, Glu580Lys, identified in a family and responsible for typical gelsolin amyloidosis phenotype. 

The mutation is localised at the interface between the G4 and G5 domains where, according to in silico analysis, is responsible for structural disruption.

The paper is well written and only a few minor corrections/spell checks are needed.

Line 57 remove "the" before "recent years"

Line 111 insert "," before "however"

Line 114 correct 'sings" into "signs"

Line 126 insert "," before "however"

Figure 4 insets especially Glu580Lys: the + and - symbols, despite being coloured differently, lack a bit of definition and are hard to read. I suggest you improve it if possible.

Line 173 maybe replace ",here," with "where". ...is reported in the insets where residue 580 and...

Lines 182-185-189 insert "red" after Congo as the dye is effectively named Congo red

Line 201 insert "also" after "was"

Line 243 remove "the" before GSN

Line 276 remove remove "the" before "recent years"

Would the authors also consider performing mass spectrometry analysis on the ex vivo material they have collected?. Mass spectrometry is indeed becoming the "gold standard" for amyloid characterisation and I believe it would benefit the paper if such data were included.

Maybe a brief description on how both Congo red and ThT staining were performed could be included.

Author Response

Dear Sir or Madam,

Thank you very much for kind suggestions and corrections of our manuscript on the new variant of gelsolin amyloidosis. We tried to respond to all suggested ideas.

Please find below the changes marked in red and responses in comments.

We are also sending you the revised version of the manuscript (file ijmsi-1067542_revised), where the corrections and additions are marked with »track changes« feature. We added some text in methods and in conclusions.

Additionaly, we corrected some minor errors in the authors names under the section 'citation' on the left side to the abstract on the first page.

We hope the answers and corrections provided will be according to reviewers wishes.

Best regards,

Maja Potrč

Reviewer 1

Line 57 remove "the" before "recent years"

Thank you, it is corrected to »In recent years, cases« and is now positioned in line 62

Line 111 insert "," before "however"

Thank you, it is corrected to »of the brain, however white matter” and is now positioned in line 117

Line 114 correct 'sings" into "signs"

Thank you, it is corrected to »signs« and is now positioned in line 120

Line 126 insert "," before "however"

Thank you, it is corrected to  »normal, however” and is now positioned in line 133

Figure 4 insets especially Glu580Lys: the + and - symbols, despite being coloured differently, lack a bit of definition and are hard to read. I suggest you improve it if possible.

A better quality image of Fig. 4 is now inserted in the article.

Line 173 maybe replace ",here," with "where". ...is reported in the insets where residue 580 and...

Thank you, it is corrected to »in the insets, where, residue 580« It is now positioned in line 182

Lines 182-185-189 insert "red" after Congo as the dye is effectively named Congo red

Thank you, it is corrected as requested and is now positioned in line 191, 194, 200

Line 201 insert "also" after "was"

Thank you, it is corrected to »phenotype was also originally« and is now positioned in line 211

Line 243 remove "the" before GSN 

Thank you, it is corrected to »Previous histopathological studies in GSN patients” and is it is now positioned in line 255

Line 276 remove »the« before »recent years«

Thank you, it is corrected to »In recent years, several GSN” and is now positioned in line 289  

Would the authors also consider performing mass spectrometry analysis on the ex vivo material they have collected?. Mass spectrometry is indeed becoming the "gold standard" for amyloid characterisation and I believe it would benefit the paper if such data were included.

This analysis is possible on FFPE tissue we have in our archives after laser microdissection. However, we do not have this technology at our institute and we do not know whether it is routinely used in Slovenia.

Maybe a brief description on how both Congo red and ThT staining were performed could be included.

Thank you for your suggestion. This is inserted in the manuscript and positioned between lines 373 and 379:

Congo red staining was performed automatically in Ventana Benchmark Special Stains stainer with Congo Red Stainning Kit (Ventana Medical Systems Inc., Tucson, AZ). For thioflavin T staining, slides were incubated in 1 % working solution of thioflavin T for 7 minutes (Sigma Aldrich, Darmstadt, Germany), rinsed in deionized water and then kept in 1 % CH3COOH for 20 minutes. Afterwards the slides were rinsed and coverslipped directly from deionized water with Dako Fluorescence Mounting Medium (DAKO, Glostrup, Denmark).

We amended Supplemental Figure 1: The labels of C and D were exchanged because the panel D was quoted prior to the panel C in the main text.

Reviewer 2 Report

The authors reported a new pathogenic variant of gelsolin, p.Glu580Lys, and its clinical and histopathological features, distinct from other gelsolin variants. In addition, the author virtually investigated the effect of the Glu to Lys mutation on the stability and structure of the protein based on the reported crystal structure. The reviewer feels that this report is valuable because it includes the detailed descriptions of the symptoms associated with this novel mutation. Furthermore, this article also assumes a small review about the pathogenic gelsolin variants reported so far: Table 2 and Figure 6 summarize the relations between the mutation positions on the sequence (and structure) and phenotypes/clinical symptoms. This information is valuable in considering the pathogenic mechanism caused by the mutation.

However, the reviewer feels that there is little discussion for the molecular basis of the pathogenicity of the mutation in this manuscript. Although the author made a putative structure of the p.Glu580Lys mutant, which offered a local destabilization due to the electrostatic repulsion, there is still a big gap between the structural destabilization and the observed symptoms. The reviewer requests the author to introduce previous reports about the changes in physicochemical or molecular properties of gelsolin (e.g., stability, amyloid fibril formation, preference of interaction to other molecules, and localization in the cell) by the other pathogenic mutations, and, based on the information, discuss a putative mechanism of the pathogenicity of the p.Glu580Lys mutation. The main readers of this journal would be interested in this point.

In addition, the reviewer listed several minor points to be modified.

  1. Throughout the manuscript: Add the panel label of each figure. For example, “Supplemental Fig. 1A” in line 73, and “Fig. 1A” in line 74. It will enhance readability to wider readers, especially who are not familiar with ophthalmological examinations.
  2. Line 96: Describe the non-abbreviated form of “OCT”: this position is the first appearance of “OCT”.
  3. Line 114: The “sings” might be “signs”.
  4. Figure 5: Add scale bars in panes A-F, too.
  5. Supplemental Figure 1: The labels of C and D should be exchanged because the panel D was quoted prior to the panel C in the main text.

Author Response

Dear Sir or Madam,

Thank you very much for kind suggestions and corrections of our manuscript on the new variant of gelsolin amyloidosis. We tried to respond to all suggested ideas.

Please find below the changes marked in grey and responses in comments.

We are also sending you the revised version of the manuscript (file ijmsi-1067542_revised), where the corrections and additions are marked with »track changes« feature. We added some text in methods and in conclusions.

Additionaly, we corrected some minor errors in the authors names under the section 'citation' on the left side to the abstract on the first page.

We hope the answers and corrections provided will be according to reviewers wishes.

Best regards,

Maja Potrč

Reviewer 2

However, the reviewer feels that there is little discussion for the molecular basis of the pathogenicity of the mutation in this manuscript. Although the author made a putative structure of the p.Glu580Lys mutant, which offered a local destabilization due to the electrostatic repulsion, there is still a big gap between the structural destabilization and the observed symptoms. The reviewer requests the author to introduce previous reports about the changes in physicochemical or molecular properties of gelsolin (e.g., stability, amyloid fibril formation, preference of interaction to other molecules, and localization in the cell) by the other pathogenic mutations, and, based on the information, discuss a putative mechanism of the pathogenicity of the p.Glu580Lys mutation. The main readers of this journal would be interested in this point.

We agree with reviewer 2, there is still a gap between impact of the novel mutation(s) on the protein and pathology but any attempt to fill this gap would be very speculative at this point. However, there are few evidence, mostly in vitro studies, suggesting alternative amyloidogenic pathways.

  1. Throughout the manuscript: Add the panel label of each figure. For example, “Supplemental Fig. 1A” in line 73, and “Fig. 1A” in line 74. It will enhance readability to wider readers, especially who are not familiar with ophthalmological examinations.

Thank you, we have added the panel labels:

  • Line 78 »but no localized scotoma (Supplemental Fig. 1A)«
  • Line 79 »white stromal corneal deposits (Fig. 1A,C)”
  • Line 81 »disc on the LE (Fig. 2A)”

  • Line 84 »daughter also had corneal problems (Fig. 1B,D, Table 1)”

  • Line 104 »RNFL around the optic nerve (Fig. 2A)”
  • Line 107 »from the RE optic disc (Supplemental Fig. 1B)”

  • Line 111 »ultrasound of the temporal arteries (Supplemental Fig. 1C)«

  • Line 113 »showed multiple calcifications in the brain nuclei (Supplemental Fig. 1D)”
  • Line 120 »was bilateral atrophy of the optic nerve (Fig. 2B)”

  • Line 133 »the slit lamp (Fig. 1B,D)”

  • Line 134 »was thinned on the macular OCT (Fig. 2C)”

  • Line 191 »for Congo red stain (Fig.5A-D) including«

  • Line 193 »T stain (Fig. 5E,F)«

  1. Line 96: Describe the non-abbreviated form of “OCT”: this position is the first appearance of “OCT”.

Thank you, it is corrected from »OCT« to »optical coherence tomography (OCT)« and is now positioned in line 102 in the main text and under description of figure 1 in the line 87

  1. Line 114: The “sings” might be “signs”.
  • Thank you, it is corrected to »signs« and is now positioned in line 120

  1. Figure 5: Add scale bars in panes A-F,

Added (Scale bars: Horizontal dimensions of the entire final figures are in µm as follows: A: 1200, B: 150, C-F:450.)

  1. Supplemental Figure 1: The labels of C and D should be exchanged because the panel D was quoted prior to the panel C in the main text.

Amended as suggested

  1. We added the following text to the conclusions, positioned between line 394-399:

Although the novel pathogenic variant share a similar clinical picture to FAF, the underlying molecular mechanism might be different. Glu580Lys mutation, as many other reviewed in this manuscript, localizes far from the second domain and unlikely leads to its destabilization and the exposure of the aberrant furin cleavage site. Recent molecular studies (Bonì et al. 2018) suggested an alternative, proteolysis-independent, mechanism, but they await in vivo validation.
